# A Novel Combination of Blood Biomarkers and Clinical Stroke Scales Facilitates Detection of Large Vessel Occlusion Ischemic Strokes

**DOI:** 10.3390/diagnostics11071137

**Published:** 2021-06-22

**Authors:** Edoardo Gaude, Barbara Nogueira, Marcos Ladreda Mochales, Sheila Graham, Sarah Smith, Lisa Shaw, Sara Graziadio, Gonzalo Ladreda Mochales, Philip Sloan, Joshua D. Bernstock, Shashank Shekhar, Toby I. Gropen, Christopher I. Price

**Affiliations:** 1Pockit Diagnostics Ltd., Cambridge CB4 2HY, UK; barbara.nogueira@pockitdx.co.uk (B.N.); marcos.ladreda@pockitdx.co.uk (M.L.M.); gonzalo.ladreda@pockitdx.co.uk (G.L.M.); 2CEPA Biobank, The Newcastle NHS Foundation Trust, Newcastle upon Tyne NE3 3HD, UK; Sheila.Graham2@nhs.net (S.G.); philip.sloan@nhs.net (P.S.); 3NovoPath Biobank, Newcastle MRC Node, Newcastle NHS Foundation Trust, Newcastle upon Tyne NE1 4LP, UK; sarah.smith153@nhs.net; 4Stroke Research Group, Population Health Sciences Institute, Newcastle University, Newcastle upon Tyne NE2 4HH, UK; lisa.shaw@ncl.ac.uk (L.S.); C.I.M.Price@newcastle.ac.uk (C.I.P.); 5Newcastle upon Tyne Hospitals NHS Foundation Trust, Newcastle upon Tyne NE2 4HH, UK; sara.graziadio@york.ac.uk; 6Brigham and Women’s Hospital, Harvard Medical School, Boston, MA 02115, USA; jbernstock@bwh.harvard.edu; 7University of Mississippi Medical Center, Jackson, MS 39216, USA; sshekhar@umc.edu; 8University of Alabama at Birmingham, Birmingham, AL 35294, USA; tgropen@uabmc.edu

**Keywords:** stroke, biomarkers, large vessel occlusions

## Abstract

Acute ischemic stroke caused by large vessel occlusions (LVOs) is a major contributor to stroke deaths and disabilities; however, identification for emergency treatment is challenging. We recruited two separate cohorts of suspected stroke patients and screened a panel of blood-derived protein biomarkers for LVO detection. Diagnostic performance was estimated by using blood biomarkers in combination with NIHSS-derived stroke severity scales. Multivariable analysis demonstrated that D-dimer (OR 16, 95% CI 5–60; *p*-value < 0.001) and GFAP (OR 0.002, 95% CI 0–0.68; *p*-value < 0.05) comprised the optimal panel for LVO detection. Combinations of D-dimer and GFAP with a number of stroke severity scales increased the number of true positives, while reducing false positives due to hemorrhage, as compared to stroke scales alone (*p*-value < 0.001). A combination of the biomarkers with FAST-ED resulted in the highest accuracy at 95% (95% CI: 87–99%), with sensitivity of 91% (95% CI: 72–99%), and specificity of 96% (95% CI: 90–99%). Diagnostic accuracy was confirmed in an independent cohort, in which accuracy was again shown to be 95% (95% CI: 87–99%), with a sensitivity of 82% (95% CI: 57–96%), and specificity of 98% (95% CI: 92–100%). Accordingly, the combination of D-dimer and GFAP with stroke scales may provide a simple and highly accurate tool for identifying LVO patients, with a potential impact on time to treatment.

## 1. Introduction

Stroke remains one of the leading causes of death and disability worldwide, with ischemic stroke caused by large vessel occlusions (LVOs) contributing disproportionally to such poor outcomes (i.e., 62% of post-stroke disability and 96% of post-stroke mortality [1]).

Acute ischemic stroke patients diagnosed with an LVO can be effectively treated via endovascular thrombectomy (EVT) [2], but this treatment is only available at comprehensive stroke centers (CSC) and/or other EVT-capable institutions. Unfortunately, the inter-hospital transfer of LVO patients from primary stroke centers to EVT-capable centers significantly delays time to treatment and leads to higher disability rates [3]. The identification of LVO patients in a pre-hospital setting (e.g., within an ambulance) would enable the transfer of LVO patients to EVT-capable centers directly [4], thereby reducing time to treatment, functional disability and/or deaths. 

While a number of studies have investigated the ability of pre-hospital stroke assessment scales to identify LVOs in the field, such measures lack the sensitivity and specificity required for triaging LVO patients with confidence [5,6]. As such, it has become clear that a more accurate diagnostic test capable of complementing these clinical assessment scores is needed. 

In line with such thinking, several research studies have investigated the ability of blood derived biomarkers to differentiate stroke from non-stroke patients and/or ischemic from hemorrhagic stroke patients [7]. In addition, a number of studies have examined blood biomarkers in an effort to determine stroke etiology [8]. Far fewer studies have investigated the association of blood biomarkers with LVO strokes specifically (i.e., as a subtype of ischemic stroke) and/or comprehensively (i.e., as a combination of different etiologic subtypes) [9,10,11]. The blood biomarkers for the identification of LVO strokes, therefore, remain to be fully elucidated/clinically validated. 

The primary aim of this study was to investigate whether the addition of blood biomarkers to stroke severity scales can improve LVO detection, compared to the use of stroke scales alone. To accomplish this, we screened a panel of biomarkers that have already been associated with stroke subtypes/etiology. The screening panel comprised D-dimer, osteopontin (OPN) and osteoprotegerin (OPG), which have previously been associated with cardioembolic and atherosclerotic stroke etiologies [8,12]; the glial fibrillary acidic protein (GFAP), an astrocyte marker that is increased in the hemorrhagic stroke subtype [7]; the von Willebrand factor (vWF), and a disintegrin and a metalloproteinase with a thrombospondin type I motif, member 13 (ADAMTS13), which are known markers of hemostasis and have been linked to the ischemic stroke subtype [13]. We then combined the best-performing biomarkers with a number of stroke severity scales and compared the accuracy for LVO detection against the use of stroke scales alone.

## 2. Materials and Methods

This study was conducted according to the Standards for Reporting Diagnostic Accuracy (STARD) guidelines [14].

### 2.1. Study Design and Sample Collection

This study was retrospective and observational in nature. The derivation cohort was collected between August 2018 and February 2020, while the validation cohort was collected between July 2020 and December 2020. Briefly, Cellular Pathology (CEPA) Biobank staff identified study eligible patients that presented to the Emergency Department (ED) at the Royal Victoria Infirmary Hospital in Newcastle upon Tyne (UK). Patients were selected based on the following criteria: (1) >18 years old; (2) evaluated in the ED for suspected stroke, as identified by ambulance paramedics, ED clinicians, and/or stroke specialist nurses; (3) <12 h from their last known well or symptom onset time; (4) reperfusion therapy had not yet been administered. As per standard clinical practice, whole venous blood was drawn at ED admission and stored at 4 °C (protected from light). Whole blood diagnostic remnants of study eligible patients were identified by biobank staff and centrifuged at 2000× *g* for 15 min at 4 °C and immediately frozen at −80 °C. Frozen samples were then transferred in one batch to Pockit Diagnostics Ltd. (Cambridge, UK) for biomarker discovery/evaluation; samples from the derivation cohort were transferred in May 2020, while samples from the validation cohort were transferred in January 2021. Informed consent was waived, and ethics approval was obtained by CEPA Biobank (NHS-HRA-North-East-Newcastle & North Tyneside 1 Research Ethics Committee, REC Reference: 17/NE/0070); all procedures performed were in accordance with prescribed institutional guidelines.

### 2.2. Clinical Data Collection

The following routine clinical data were collected for eligible patients: demographics (i.e., age, sex), clinical characteristics (i.e., blood pressure, pulse, atrial fibrillation, hypertension), clinical laboratory results (i.e., complete blood count (CBC), biochemistry, blood lipids), the NIH Stroke Scale (NIHSS) score at presentation, last known well or symptom onset time, blood withdrawal time, imaging findings within 1 h, and final clinical diagnosis. Data were provided to Pockit Diagnostics Ltd. in a blinded fashion.

### 2.3. Assigning a Diagnostic Category

The routine clinical data collected above were ultimately used to assign patients to the following diagnostic categories: LVO, non-LVO ischemic stroke, hemorrhagic stroke, transient ischemic attack (TIA), or stroke mimic condition. Transient ischemic attack (TIA) and stroke mimic were assigned based on the opinion of a clinician with clinical expertise. Hemorrhagic stroke was assigned based on the presence of extravascular parenchymal blood on cranial imaging. LVO required computed tomography angiography (CTA) evidence, as confirmed by a neuroradiology report. Those remaining patients to whom a stroke specialist assigned a diagnosis of ischemic stroke were categorized as either non-LVO or not classifiable as per the following: -non-LVO if CTA had been undertaken and LVO was not present or if a CTA had not been undertaken upon admission, but the NIHSS score was <5. The latter was a pragmatic threshold reflecting a low likelihood of LVO [15];-not classifiable if CTA had not been undertaken and NIHSS score on admission was >4.

### 2.4. Derivation of Stroke Scales from NIHSS Score

FAST score was calculated by assigning 1 point for the presence of facial paresis (NIHSS item 4), 1 point for any arm weakness (NIHSS item 5a/b), and 1 point for any speech impairment (NIHSS item 9). FAST-ED was calculated as described by Lima et al. [16], RACE score was calculated as described by Perez de la Ossa [17], C-STAT was calculated as described by Katz et al. [18], and EMSA was calculated as described by Gropen et al. [19]. 

### 2.5. Measurement of Blood Biomarkers

Samples from the derivation cohort were tested in June 2020 and samples of the validation cohort were tested in January 2021. Plasma biomarkers were measured using commercially available enzyme-linked immunosorbent assays (ELISA), as per the manufacturer’s instructions. ELISA kits and/or matched antibody pairs were purchased from Abcam (Cambridge, UK): D-dimer (product number: ab196269), OPN (product number: ab100618), OPG (product number: ab100617), GFAP (product number: ab222279), vWF (product number: ab223864), and ADAMTS13 (product number: ab234559). Plasma sample dilutions for each biomarker sample were as follows: D-dimer (1:80), OPN (1:2), OPG (1:6), GFAP (1:2), vWF (1:4000), ADAMTS13 (1:800). Samples were measured in a randomized order. All samples were analyzed in duplicate, and the mean value was used for quantification. All readings were performed with a Multiskan™ FC spectrophotometer (Thermo-Fisher Scientific, Catalog Number 51119000). For all biomarkers, the average coefficient of variation was <10%. GraphPad Prism version 8.4.3 was used for biomarker quantification/analysis.

### 2.6. Statistical Analyses

The proposed intended use of the blood biomarkers is to rule in patients with LVO to redirect patients to EVT-capable centers from a population of suspected stroke; thus, we powered the study on specificity. We assumed a specificity of 97% with a minimal acceptable specificity of 90%, a two-tailed 5% type I error rate (α) and 90% power (β). With these assumptions, we originally calculated a sample size of 161 suspected stroke patients, with 45 LVO cases. Recruitment was interrupted by the COVID-19 pandemic and, after exclusion of samples with unknown LVO diagnosis, we obtained a workable sample set of 128 suspected stroke patients, with 23 LVO cases. Performing a power calculation based on the available sample size and the observed LVO prevalence in our cohort (18% instead of the assumed 28%) indicated that we would still have 86% power to achieve our goal of demonstrating a specificity of greater than 90%. This initial cohort is referred to as the “derivation cohort”. After the lifting of COVID-19 clinical research restrictions, a further sample of 119 suspected stroke patients was recruited, which yielded a final workable sample set of 111 patients with 17 LVOs. This latter cohort is referred to as the “validation cohort”.

Symptom onset to blood collection time (OBT) was calculated using the last known well/onset time and documented time of the blood draw. If patients were reported to have woken up with symptoms, the time of last known well was assumed to be midnight of the previous day. Where a range for onset time/last known well time was provided, the first-time value was used to calculate OBT.

To compare the levels of blood-derived biomarkers and clinical variables in both LVO and non-LVO patients, the distribution normality of continuous variables was assessed using the Shapiro-Wilk test and the following: (i) for normally distributed variables, Welch’s *t*-test and mean ± standard deviation (SD) were used or (ii) for non-normally distributed variables, the Wilcoxon-Mann-Whitney U test, median and interquartile range (IQR) were used. Categorical variables were assessed via Pearson’s chi-square test. When >10 variables were tested at the same time, multiple hypothesis correction was performed using the Benjamini-Hochberg method.

To identify the optimal panel of blood biomarkers for LVO prediction, we employed multivariate logistic regression with diagnosis (LVO vs. non-LVO) as the outcome variable and the plasma levels of D-dimer, GFAP, OPN, OPG, vWF, and ADAMTS13 as the exploratory variables. Bidirectional stepwise elimination based on Akaike information criterion (AIC) levels was used for model selection. Linearity between predictors and the outcome measure was assessed through logarithmic and quadratic transformation; transformations were selected based on the AIC.

To investigate whether the addition of blood-based biomarkers improved the accuracy of clinical stroke severity scales for LVO identification, we used a second multivariate logistic regression with diagnosis as the outcome variable and the optimal biomarker panel and one of the stroke severity scales (FAST, FAST-ED, RACE, C-STAT, or EMSA) as the exploratory variables. 

To assess the goodness of fit of our blood biomarker panel and the stroke scales, the likelihood ratio test (LR) and AIC were used. The area under the receiver operating characteristic curve (AUC) with 95% CIs was used as a measure of discrimination. For each model, the cut-off point was selected by maximizing the specificity for LVO prediction, in line with our power calculation.

The cut-off points were estimated using the dataset from the derivation cohort and were subsequently tested on the validation cohort for diagnostic accuracy.

At selected cut-off points, accuracy, sensitivity, specificity, positive likelihood ratio (LR+), and negative likelihood ratio (LR−) were also evaluated. Corrections for optimistic predictions was performed through bootstrapping with 2000 resamples and presented with confidence intervals (CI).

All analyses were performed with R version 3.6.2 with the help of RStudio version 1.2.5033 by using the packages MKmisc, nnet, pROC, caret, tidyverse, oddsratio, lmtest, and OptimalCutpoints.

## 3. Results

### 3.1. Derivation Cohort

Data from 170 patients with suspected strokes were collected in our derivation cohort. Blood samples from 19 patients were utilized for initial immunoassay testing and excluded from further use. Data from a further 23 patients could not be categorized as LVO or non-LVO and were thus excluded from the final analysis. The final cohort of 128 suspected stroke patients was categorized as follows (Appendix A): LVO ischemic strokes (*n* = 23, 18%), non-LVO ischemic strokes (*n* = 42, 33%), hemorrhagic strokes (*n* = 16, 12.5%), stroke mimics (*n* = 31, 24%), and transient ischemic attacks (*n* = 16, 12.5%). The stroke mimic diagnoses were as follows: anemia (3%), anxiety (10%), Bell’s palsy (7%), delirium (3%), dementia (3%), depression (3%), dysphasia (3%), metastatic cancer (3%), migraine (16%), seizure (13%), syncope (13%), vertigo (13%), and undetermined (10%).

The clinical characteristics for LVO and all non-LVO patients from the derivation cohort are reported in Table 1. In our cohort, we found significant differences between LVO and all non-LVO with regard to the NIHSS score (18 ± 9 and 3 ± 5, *p*-value < 0.001), the presence of atrial fibrillation (52% and 10%, *p*-value < 0.001), and systolic blood pressure (140 ± 22 and 157 ± 29 mmHg, *p*-value = 0.03). Of note, age, sex, or time from stroke onset to blood collection (OBT) were not different between LVO and non-LVO patients.

### 3.2. Blood Biomarker Panel

We then compared the plasma levels of D-dimer, OPN, OPG, vWF, and ADAMTS13 in LVO and non-LVO patients; the plasma levels of GFAP were compared in hemorrhagic vs. ischemic stroke and non-stroke patients (Figure 1 and Appendix A). We found statistically significant differences between LVO and all non-LVO for the following blood biomarkers: D-dimer (1.3 ± 2.0 and 0.4 ± 0.4 µg/mL, *p*-value < 0.001); OPN (1.7 ± 1.1 and 1.2 ± 1.1 ng/mL, *p*-value = 0.02); and OPG (125 ± 60 and 96 ± 54 pg/mL, *p*-value = 0.01). In addition, GFAP was significantly increased in hemorrhagic stroke, as compared to all the other suspected stroke patients (1043 ± 2581 and 66 ± 130 pg/mL, *p*-value < 0.05; Figure 1b).

The levels of D-dimer, OPN, OPG, and ADAMTS13 were log transformed, while the levels of vWF underwent quadratic transformation; no transformation was applied to the GFAP levels. Among the six measured biomarkers, the bi-directional stepwise feature selection identified D-dimer (OR 16, 95% CI 5–60; *p*-value < 0.001) and GFAP (OR 0.002, 95% CI 0–0.68; *p*-value < 0.05; Appendix A) as the optimal parsimonious panel for LVO identification.

### 3.3. Combination of Blood Biomarkers and Clinical Stroke Scales

In the derivation cohort, all stroke severity scales were significantly increased in LVO, as compared to non-LVO patients (*p*-values < 0.001; Appendix A).

We combined D-dimer and GFAP with each stroke severity scale into multivariable logistic regression models. The models built with the combination of blood biomarkers and stroke scales had lower AIC values, higher AUC and significant LR test *p*-values, as compared to the stroke scales alone (Table 2 and Figure 2). D-dimer and GFAP had highly significant *p*-values in the combined models (Appendix A), indicating that both biomarkers significantly contribute to LVO detection, regardless of what stroke scales they are combined with. D-dimer and GFAP significantly improved the accuracy of clinical scales, such that all combinations achieved high performance with regard to LVO prediction (Table 3); of note, a combination with FAST-ED resulted in the highest accuracy of 95% (95% CI 87–99%), an LR^+^ of 23 (95% CI 9–60), an LR^−^ of 0.09 (95% CI 0.02–0.34), a sensitivity of 91% (95% CI 72–99%), and a specificity of 96% (95% CI 90–99%; Table 3). Overall, the addition of D-dimer and GFAP to stroke scales allowed us to increase the number of true positives, while reducing the number of false positives due to hemorrhage (Appendix A).

### 3.4. Validation Cohort

Data from an additional 119 patients with suspected stroke were collected in our validation cohort. Data from eight patients could not be categorized, and they were, therefore, excluded from analysis. The remaining 111 patients were categorized as follows: LVO ischemic strokes (*n* = 17, 15%), non-LVO ischemic strokes (*n* = 43, 39%), hemorrhagic strokes (*n* = 9, 8%), stroke mimics (*n* = 30, 27%), and transient ischemic attacks (*n* = 12, 11%). The stroke mimic diagnoses were as follows: anisocoria (3.3%), Bell’s palsy (3.3%), cancer (3.3%), delirium (6.7%), functional neurological disorder (13.3%), migraine (30%), seizure (10%), syncope (6.7%), and undetermined (23%). 

As with our derivation cohort, we again found significant differences between LVO and all non-LVO patients with regard to the NIHSS score (23 ± 10 and 4 ± 6, *p*-value < 0.001) and the presence of atrial fibrillation (41% and 17%, *p*-value < 0.01). No differences in age, sex, or OBT were found between LVO and non-LVO patients in the validation cohort.

In our validation cohort, the plasma levels of D-dimer were significantly increased in LVO, as compared to non-LVO patients (1.6 ± 1.7 and 0.9 ± 1.4 µg/mL, *p*-value < 0.001; Figure 3). In addition, GFAP was found to be significantly increased in hemorrhagic stroke patients, as compared to ischemic stroke and all non-stroke patients (434 ± 566 and 95 ± 226 pg/mL, *p*-value < 0.01; Figure 3). 

### 3.5. Validation of Diagnostic Accuracy

In the validation cohort, all stroke severity scales were significantly increased in LVO, as compared to non-LVO patients (*p*-values < 0.001; Appendix A). We tested the diagnostic accuracy of the derived logistic model cut-off points on the validation cohort. In line with the results obtained using the derivation cohort, the combination of D-dimer and GFAP with any stroke severity scale increased the accuracy for LVO detection, when compared to stroke scales alone (Table 4). The model algorithm built with D-dimer, GFAP, and FAST-ED achieved an accuracy of 95% (95% CI 87–99%), a sensitivity of 82% (95% CI 57–96%), a specificity of 98% (95% CI 92–100%), an LR+ of 37 (95% CI 9.2–148), and an LR− of 0.18 (95% CI 0.06–0.5).

## 4. Discussion

Herein, we have demonstrated that a biomarker panel composed of both D-dimer and GFAP, when combined with clinical stroke scales, may serve as a valuable tool for the specific identification of stroke patients with LVOs. Importantly, we have also validated and confirmed the diagnostic accuracy of our findings in an independent patient cohort.

In a recent study by Lopez-Cancio and colleagues, D-dimer has been associated with LVOs [11]. The authors found that a cut-off point for D-dimer at 1664.15 ng/mL, together with a NIHSS score ≥ 10 was capable of detecting LVOs with a specificity of 93% and a sensitivity of 35%. While we observed a similar diagnostic specificity for LVO, we found that combining D-dimer and GFAP with stroke severity scales allowed us to detect LVO with greater sensitivity, compared to the study by Lopez-Cancio et al. This difference may be due to the addition of GFAP to D-dimer, and/or to the use of different stroke severity scales. 

Previous work has shown that GFAP levels are elevated in hemorrhagic stroke patients, as compared to ischemic strokes, stroke mimics, and/or TIAs [20,21]. Our findings confirm such reports, as we did indeed see higher plasma levels of GFAP in hemorrhagic patients from both our cohorts. To our knowledge, no studies have addressed the role of GFAP in the identification of LVO patients. Our study demonstrates that, when measured with D-dimer, GFAP can significantly improve LVO identification by ruling out hemorrhagic patients whose clinical stroke scales often suggest the presence of LVOs. Of note, the proportion of hemorrhagic strokes observed in our study was significant (8–12.5%), and is in line with the previously observed prevalence for this type of stroke (8–15%) [22]. This clearly indicates the importance of a tool capable of ruling out hemorrhage in clinical scenarios where a high specificity for LVO detection is required.

We estimated the diagnostic performance of a number of validated stroke severity scales for LVO prediction and observed that their overall accuracy was higher, as compared to a litany of previous studies [16,17,18,19,23,24]. This may be due to the derivation of our employed stroke scales from the NIHSS score, which was performed by ED clinicians; such a finding may be particularly relevant for the FAST-ED and RACE, which are known to be more complex as compared to FAST or EMSA [19]. 

It is prudent to note that other studies have shown that the combination of clinical variables with blood biomarkers can improve stroke diagnosis. Brouns and colleagues reported that the addition of the Oxford Community Stroke Project (OCSP) classification to D-dimer measurements was capable of improving the accuracy of lacunar stroke identification from 88 to 98% [25]. Lopez-Cancio et al. showed that combining D-dimer with the NIHSS score and the presence of atrial fibrillation, resulted in a better diagnostic accuracy for LVO, as compared to the biomarker alone [11]. In line with such evidence, here we demonstrate that combining biomarkers and stroke severity scales does indeed lead to a higher predictive ability for LVOs, compared to the use of stroke scales alone. Interestingly, we observed that the degree of increase in predictive ability varied when the biomarkers were combined with different NIHSS-constructed scales. This could be due to random noise within the dataset for different NIHSS items and/or to ceiling effects with regard to the high AUC. 

Each of the NIHSS-derived scales has advantages and disadvantages; the combination of the biomarkers with FAST-ED resulted in the highest diagnostic performance for LVO overall. Nevertheless, the collection of FAST-ED is more complex compared to other stroke scales, such as FAST or EMSA [19], which may be preferred in the pre-hospital setting. We showed that the combination of D-dimer and GFAP with FAST or EMSA achieved a sufficient LVO prediction (AUC = 93%) and may offer a valuable tool for the pre-hospital setting. 

High levels of specificity are required when identifying LVO patients in the field, in order to bypass the nearest stroke center and transfer patients to an EVT-capable center [5,26]. Critically, our findings indicate that combining D-dimer and GFAP with stroke severity scales has the potential to provide the level of diagnostic performance needed to safely triage LVO patients. 

In our study, we measured blood biomarkers with standard ELISA immunoassays, which require several hours and expert laboratory staff, and are therefore unsuitable for implementation in acute stroke triage. The development of a point-of-care device able to measure D-dimer and GFAP rapidly (e.g., <10 min), as well as to automatically combine field-collected stroke severity scales in an algorithm, would be required to implement our diagnostic strategy in the clinic. Previous studies have evaluated the performance of Biosite’s Triage Stroke Panel point-of-care blood test for the detection of stroke subtypes in the clinical environment [27,28]. Glickman et al. found that the measurement of the C-reactive protein, matrix metalloproteinase 9, and protein S-100b added discriminative power for ischemic stroke, compared to using the admission NIHSS score alone in the emergency department [27]. Sibon and colleagues observed that the Triage Stroke Panel had similar accuracy for ischemic stroke detection compared to a well-trained triage nurse in a stroke center [28]. These results suggest that point-of-care blood tests could complement current clinical practice and aid in stroke patient triage.

We note that our study had several practical limitations. We used blood samples derived from routine clinical tests that were stored in the dark at +4 °C before processing. Although previous studies have shown the stability of blood biomarkers up to 240 h at +4 °C [29,30], we cannot exclude the idea that some of our negative results were due in part to protein degradation. The biomarker measurement was performed using standard laboratory immunoassays, which are inherently variable, and our findings will therefore require validation. The symptom onset times were obtained from medical records by non-specialists and there may be some inaccuracies, but it is likely that the vast majority of patients presented within 12 h of onset. Categorization into LVO and non-LVO patients was based on routinely available clinical and imaging findings and included a pragmatic decision for handling ischemic stroke patients in cases where a CTA was not available. This pragmatic method may have led to the loss of LVO cases from our cohorts. Moreover, considering that 10% of LVO have an NIHSS score < 5 [31], we may have incorrectly assigned mild strokes as non-LVOs. Future studies should include a CTA for all ischemic stroke patients, thereby allowing for the independent adjudication of LVO. Some of the confidence intervals estimated in our study had a wide range. This effect could be due to the small sample size and these results should be interpreted with caution. Previous studies have highlighted the relationship between anticoagulant medication and plasma levels of D-dimer [32,33]. Information on patient’s medication was not collected in our study and an association between use of anticoagulants and D-dimer cannot be excluded. Finally, our study was retrospective and observational in nature; future studies, such as our ongoing prospective Phase 1, will be required to definitely demonstrate the clinical utility of our biomarkers in combination with stroke scales for the identification of LVO. 

In conclusion, our study strongly suggests that the combination of D-dimer and GFAP with stroke severity scales in a diagnostic algorithm may offer a valuable tool for the early identification of LVO stroke patients. 

## Figures and Tables

**Figure 1 diagnostics-11-01137-f001:**
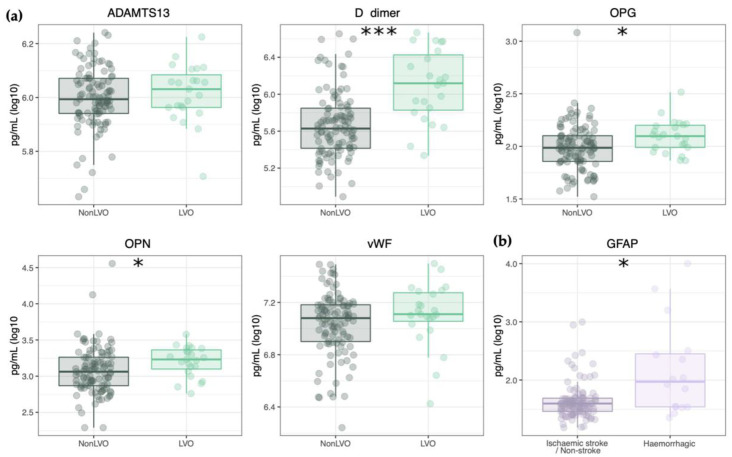
Plasma biomarker concentrations in the derivation cohort. (**a**) Plasma concentrations of blood biomarkers measured in LVO and non-LVO stroke patients; (**b**) Plasma concentrations of GFAP measured in ischemic stroke and non-stroke vs. hemorrhagic stroke patients. * and *** indicate *p*-value < 0.05 and *p*-value < 0.001, respectively. Data are shown as mean ± standard deviation.

**Figure 2 diagnostics-11-01137-f002:**
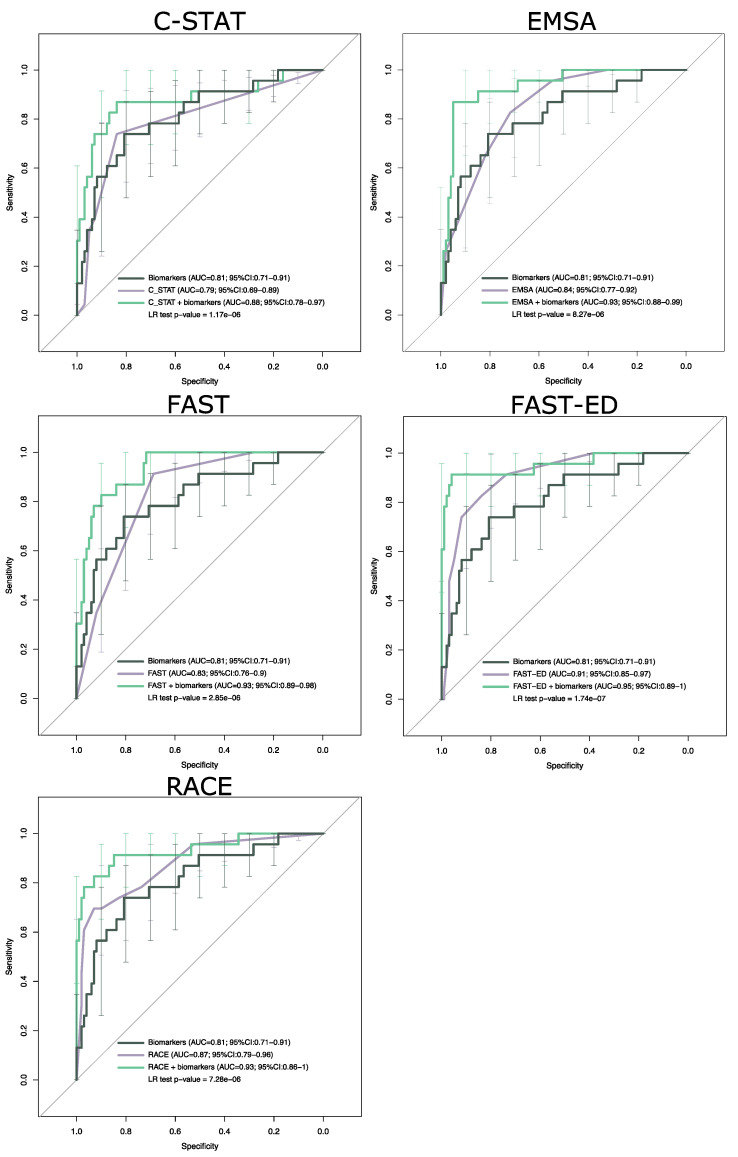
ROC curves of biomarker model and stroke scales models with or without the addition of biomarkers. Error bars indicate sensitivity confidence intervals. AUCs with 95% CIs are shown for each model. Likelihood ratio (LR) *p*-values for the comparison of each stroke scale alone each combined model (scale + biomarkers) are shown.

**Figure 3 diagnostics-11-01137-f003:**
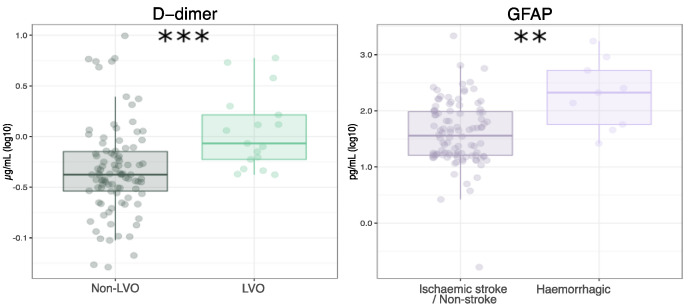
Plasma concentrations of D-dimer and GFAP in the validation cohort. Plasma concentrations of D-dimer measured in LVO and non-LVO stroke patients (**left**) and plasma concentrations of GFAP measured in ischemic stroke and non-stroke vs. hemorrhagic stroke patients (**right**). ** and *** indicate *p*-value < 0.01 and *p*-value < 0.001, respectively. Data are shown as mean ± standard deviation.

**Table 1 diagnostics-11-01137-t001:** Univariate analysis of clinical variables in LVO and non-LVO stroke patients.

Clinical Characteristics	LVO Mean (SD ^1^)	Non-LVO Mean (SD ^1^)	*p*-Value
Sex (F/M)	11/12	60/45	0.90
Age	75 (13)	77 (21)	1
Atrial fibrillation (% yes)	52	10	<0.001
Systolic blood pressure	140 (22)	157 (29)	0.03
Diastolic blood pressure	80 (24)	83 (18)	0.85
Hypertension (% yes)	70	58	0.85
APTT	29 (5)	30 (6)	0.87
Hematocrit	0.38 (0.07)	0.41 (0.06)	0.78
Prothrombin time	12 (0)	12 (1)	0.81
Fibrinogen	4.9 (1.3)	4.7 (1.1)	0.87
Platelet count	227 (76)	247 (85)	0.85
Glucose	6.8 (2.8)	6.0 (2.1)	0.21
NIHSS score	18 (9)	3 (5)	<0.001
OBT (min) ^2^	155 (179)	161 (154)	1

^1^ SD: Standard deviation; ^2^ OBT: Stroke onset to blood collection time.

**Table 2 diagnostics-11-01137-t002:** Model comparisons.

Model	AIC ^1^	AUC ^2^	LR ^3^ (df), *p*-Value
C-STAT	102.36	79 (72–86)	-
C-STAT + D-dimer + GFAP	79.29	88 (81–94)	27.3 (4), <0.001
EMSA	89.75	84 (79–89)	-
EMSA + D-dimer + GFAP	70.34	93 (89–97)	23.4 (4), <0.001
FAST	93.35	83 (78–88)	-
FAST + D-dimer + GFAP	71.82	93 (90–97)	25.5 (4), <0.001
FAST-ED	78.25	91 (86–95)	-
FAST-ED + D-dimer + GFAP	51.12	95 (91–100)	31.1 (4), <0.001
RACE	78.87	87 (82–93)	-
RACE + D-dimer + GFAP	59.21	93 (89–98)	23.7 (4), <0.001

^1^ AIC: Akaike Information Criterion; ^2^ AUC: Area under the receiver operating characteristic curve, presented with 95% confidence intervals; ^3^ LR: Likelihood ratio test, presented with degree of freedom (df) and *p*-values.

**Table 3 diagnostics-11-01137-t003:** Internal validation of diagnostic accuracy.

Model	Accuracy	Sensitivity	Specificity	LR+	LR−
C-STAT	84 (79–88)	35 (19–49)	95 (92–98)	8 (3–20)	0.69 (0.54–0.85)
C-STAT + D-dimer + GFAP	89 (79–96)	74 (52–90)	93 (86–97)	10 (5–22)	0.28 (0.14–0.56)
EMSA	79 (73–84)	65 (50–80)	82 (76–87)	3.7 (2.5–5.3)	0.42 (0.24–0.6)
EMSA + D-dimer + GFAP	93 (84–98)	87 (66–97)	95 (89–98)	17 (7–41)	0.14 (0.05–0.39)
FAST	73 (67–78)	91 (83–96)	69 (62–75)	2.9 (2.3–3.7)	0.13 (0.06–0.25)
FAST + D-dimer + GFAP	90 (80–96)	78 (56–93)	93 (86–97)	11 (5–23)	0.23 (0.11–0.51)
FAST-ED	84 (78–88)	83 (71–95)	84 (79–89)	5.3 (3.6–7.6)	0.21 (0.06–0.35)
FAST-ED + D-dimer + GFAP	95 (87–99)	91 (72–99)	96 (90–99)	23 (9–60)	0.09 (0.02–0.34)
RACE	86 (82–91)	70 (56–85)	90 (86–94)	7.3 (4.6–12.4)	0.33 (0.17–0.49)
RACE + D-dimer + GFAP	91 (81–97)	83 (61–95)	93 (86–97)	12 (6–24)	0.19 (0.08–0.46)

All diagnostic measures are presented with 95% confidence intervals.

**Table 4 diagnostics-11-01137-t004:** External validation of diagnostic accuracy.

Model	Accuracy	Sensitivity	Specificity	LR+	LR−
C-STAT	89 (85–93)	65 (47–82)	93 (90–97)	11 (6–20)	0.38 (0.2–0.57)
C-STAT + D-dimer + GFAP	89 (78–96)	71 (44–90)	92 (85–97)	9 (4–20)	0.32 (0.15–0.67)
EMSA	69 (63–75)	100 (100–100)	63 (56–70)	3 (2–3)	0 (0–0)
EMSA + D-dimer + GFAP	90 (79–96)	88 (64–99)	90 (82–95)	9 (5–17)	0.13 (0.04–0.48)
FAST	89 (85–93)	65 (47–81)	93 (90–97)	11 (6–23)	0.38 (0.2–0.56)
FAST + D-dimer + GFAP	88 (77–95)	71 (44–90)	91 (83–96)	8 (4–16)	0.32 (0.15–0.68)
FAST-ED	89 (85–93)	88 (77–100)	89 (84–93)	8 (6–14)	0.13 (0–0.26)
FAST-ED + D-dimer + GFAP	95 (87–99)	82 (57–96)	98 (92–100)	37 (9–148)	0.18 (0.06–0.5)
RACE	87 (82–92)	88 (77–100)	87 (81–92)	7 (5–11)	0.14 (0–0.26)
RACE + D-dimer + GFAP	91 (80–97)	82 (57–96)	92 (85–97)	11 (5–22)	0.19 (0.07–0.54)

All diagnostic measures are presented with 95% confidence intervals.

## Data Availability

The data presented in this study are available on request from the corresponding author. The data are not publicly available due to privacy.

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
