# Peer review of "A Novel Combination of Blood Biomarkers and Clinical Stroke Scales Facilitates Detection of Large Vessel Occlusion Ischemic Strokes"

_diagnostics, 2021, doi:10.3390/diagnostics11071137_

Round 1
Reviewer 1 Report
This is a well written and well referenced manuscript. Statistical analysis fully supports data execution and the conclusion section is fully supported by the data that has been generated in this study. This manuscript can be accepted in its present form and I have no objection to the publication of this work to the Diagnostics journal.
Author Response
We want to thank the reviewer for taking the time for this review and for appreciating our work.
Reviewer 2 Report
Thank you for giving me an opportunity to read this very interesting and well-designed study. It certainly deserves to be published. The material and methods section was described in details, together with the construct of multivariate models and their validation.
I have only few minor comments:
- Confidence intervals, especially according to FAST-ED + biomarkers in the Supplementary Table 3 are broad (23.43 to 209.78), probably due to small sample size. I would recommend to add one sentence in the limitation section that, therefore, the results need to be interpreted with caution.
- There is no information on concomitant anticoagulant use and glucose levels which could potentially affect D-dimer levels. It is especially important with regard to anticoagulant use as 50% of patients with LVO had concomitant AF. Do the authors have the access to data on concomitant treatment and were there any significant differences in anticoagulant use and glucose levels between LVO and non-LVO groups? If not, I would perceive this as possible limitation of the study.
- In the section 3.5, the authors stated “We tested the diagnostic accuracy of the derived cut-off points on the validation cohort”. What exactly were the cut-off values of D-dimer and GFAP?
- As authors stated, blood biomarkers measurements “required several hours”. Are there any studies in which fast blood tests on possible stroke biomarkers were evaluated in clinical practice?
- In the first paragraph of the 3.2 section, there was a mistake in the D-dimer level (it was different than the one found in the Supplementary Table 1).
- In the third paragraph of the 3.4 section, the D-dimer level was shown in ug/ml whereas in the Figure 3 pg/ml was used. To be more transparent, I would choose to show values of D-dimer as ug/ml.
- Instead of using term “gender”, I would prefer to use “sex”.
